# Capsiate Intake with Exercise Training Additively Reduces Fat Deposition in Mice on a High-Fat Diet, but Not without Exercise Training

**DOI:** 10.3390/ijms22020769

**Published:** 2021-01-14

**Authors:** Deunsol Hwang, Jong-Beom Seo, Hun-Young Park, Jisu Kim, Kiwon Lim

**Affiliations:** 1Physical Activity and Performance Institute (PAPI), Konkuk University, Gwangjin-gu, Seoul 05029, Korea; hds49@konkuk.ac.kr (D.H.); parkhy1980@konkuk.ac.kr (H.-Y.P.); kimpro@konkuk.ac.kr (J.K.); 2Department of Sports Medicine and Science in Graduated School, Konkuk University, Gwangjin-gu, Seoul 05029, Korea; syk1528@konkuk.ac.kr; 3Department of Physical Education, Konkuk University, Gwangjin-gu, Seoul 05029, Korea

**Keywords:** exercise training, capsiate, capsaicin, obesity, metabolism, energy expenditure, abdominal fat, adrenoceptor, skeletal muscle, adipose tissue

## Abstract

While exercise training (ET) is an efficient strategy to manage obesity, it is recommended with a dietary plan to maximize the antiobesity functions owing to a compensational increase in energy intake. Capsiate is a notable bioactive compound for managing obesity owing to its capacity to increase energy expenditure. We aimed to examine whether the antiobesity effects of ET can be further enhanced by capsiate intake (CI) and determine its effects on resting energy expenditure and metabolic molecules. Mice were randomly divided into four groups (*n* = 8 per group) and fed high-fat diet. Mild-intensity treadmill ET was conducted five times/week; capsiate (10 mg/kg) was orally administered daily. After 8 weeks, resting metabolic rate and metabolic molecules were analyzed. ET with CI additively reduced the abdominal fat rate by 18% and solely upregulated beta-3-adrenoceptors in adipose tissue (*p* = 0.013) but did not affect the metabolic molecules in skeletal muscles. Surprisingly, CI without ET significantly increased the abdominal fat rate (*p* = 0.001) and reduced energy expenditure by 9%. Therefore, capsiate could be a candidate compound for maximizing the antiobesity effects of ET by upregulating beta-3-adrenoceptors in adipose tissue, but CI without ET may not be beneficial in managing obesity.

## 1. Introduction

Obesity causes an endocrine imbalance that can lead to various metabolic disorders such as cardiovascular disease and type 2 diabetes [1]. This is because excess adipose tissue abnormally generates a large amount of cytokines and bioactive mediators, namely leptin and interleukin-6 [2]. Eventually, obesity increases the risk of mortality [3] and is a persistent global health issue. Thus, considerable efforts are being made globally to reduce the incidence of obesity and obesity-associated disorders.

Exercise is one of the primary and most efficient ways to manage obesity. Exercise training (ET) is not only an energy-burning activity, but it also enhances the fat oxidation capacity [4], fat-free mass [5], and energy metabolism levels at rest [6], eventually leading to an negative energy balance and weight loss. Although ET is known to be an obvious antiobesity activity, exercise-induced weight loss is often lower than that predicted by the total expenditure of calories [7,8]. A recent study found that this discrepancy, known as “weight compensation,” results primarily from a concomitant increase in appetite and subsequent increased energy intake, especially in response to long-term exercise [9]. This phenomenon has also been reported in rodents [10]. It indicates that a dietary strategy may be necessary to maximize the antiobesity function of exercise especially in the long-term [9,11]. A typical dietary strategy used in the management of obesity is the intake of bioactive compounds to increase the level of energy expenditure [12,13].

Capsiate, first isolated from the “CH-19 Sweet pepper,” is a capsaicin analog [14] (Figure 1) and a notable bioactive compound [15]. While capsaicin has a strong, spicy taste and ingestion of large amounts can lead to gastritis [16], capsiate is not pungent [17]. It can also enhance energy expenditure to an extent equal to that of capsaicin [18,19]. Owing to these reasons, capsiate was chosen and studied as a dietary compound over capsaicin.

Capsiate enhances energy metabolism mainly via the activation of the sympathetic nervous system (SNS) [20,21]. Subsequently, blood noradrenaline and adrenaline concentration levels are enhanced [21,22], and the resting oxygen uptake is increased [19,22] upon the acute intake of capsiate. Moreover, capsiate intake (CI) for more than 2 weeks significantly reduced body weight, body fat percentage [23], and abdominal fat [19,24] both in humans and in a rodent model. Thus, capsiate has been considered a possible dietary supplement to ameliorate obesity [15].

Collectively, the results of these previous studies highlight the benefits of ET and CI in preventing obesity. However, only one study investigated the combinatorial effects of exercise and capsiate, but it had its limitations as the total amount of CI and exercise volume were not strictly controlled [25]. Meanwhile, the antiobesity effects of exercise in combination with capsaicin are well documented [26,27], suggesting the probable benefits of combination treatment with exercise and capsiate in ameliorating obesity. Therefore, the aim of this study was to examine whether the antiobesity effects of ET can be further enhanced by CI through a change in resting energy expenditure in diet-induced obese mice as well as its effects on metabolism-associated molecules and key enzymes in the blood, adipose tissue, and skeletal muscles. Despite the fact that skeletal muscles are the most critical determinants of the resting energy metabolism rate [28], no study has examined the effects of ET with CI on metabolism-related molecules in skeletal muscles.

## 2. Results

### 2.1. Body Weight, Food Intake, and Feed Efficiency Ratio

At the beginning of the experiment, there was no significant difference in body weight (BW) among the groups. After 8 weeks of mild-intensity exercise and CI together with a high-fat diet, the post-BW of the exercise-trained with CI group (EXE+CAP) clearly tended to be lower than that of the exercise-trained vehicle group (EXE), but there was no statistical difference (Table 1). However, post-BW of EXE+CAP was significantly lower than that of the sedentary with CI group (CAP), and there was no difference between the EXE and sedentary vehicle groups (CON). Subsequently, the body weight gain (BWG) of EXE+CAP tended to be lower than that of EXE and significantly lower than that of CAP, but there was no difference in BWG between EXE and CON (Table 1).

Notably, the food intake (FI) of EXE+CAP was significantly higher than that of EXE (Table 1). Furthermore, the FI of EXE+CAP or EXE was significantly higher than that of CAP or CON, respectively. Overall, the feed efficiency ratio (FER) of EXE+CAP was clearly the lowest compared to the other groups and was significantly lower than that of CAP. However, there was no difference between the EXE and CON groups (Table 1).

Thus, the antiobesity effects of ET can be further enhanced by CI in diet-induced obese mice.

### 2.2. The Proportion of Abdominal Fat

Similar to the above results, the proportions of epididymal, perirenal, mesenteric, and total abdominal fat of EXE+CAP clearly tended to be lower than those of EXE and also exhibited the lowest levels compared to the other groups (Table 1). Additionally, the fat proportions of EXE+CAP were significantly lower than those of CAP, but there was no difference between EXE and CON. Thus, ET with CI has additive effects on preventing obesity.

Additionally, interestingly, there were unexpected results in the sedentary groups. The proportion of abdominal fat of CAP was significantly higher than that of CON (Table 1).

### 2.3. Metabolism-Related Molecules in Blood

The concentration of leptin of EXE+CAP was significantly lower than that of EXE. Furthermore, leptin concentration of EXE+CAP was significantly lower than that of CAP, but there was no difference between CAP and CON (Table 2). Additionally, interestingly, the leptin concentration of CAP was significantly higher than that of CON. These results were in line with our results of the proportion of adipose tissue (Table 1).

The concentration of noradrenaline of CAP was most upregulated compared to that in the other groups. The noradrenaline level of CAP was significantly higher than that of CON (Table 2). Additionally, the noradrenaline concentration of EXE+CAP was significantly lower than that of CAP, but there was no difference between EXE and CON. The concentrations of glucose and insulin were not significantly different among the groups (Table 2).

Thus, ET combined with CI can further reduce the deposition of abdominal fat despite increasing energy intake in diet-induced obese mice. On the contrary, we found that CI without exercise could increase the abdominal fat rate. This finding is contrary to those of most previous studies that demonstrated the antiobesity effects of capsiate. From the perspective of energy balance, an analysis of the resting metabolic rate (RMR) might be able to explain these results.

### 2.4. Resting Metabolic Rate

From the results of energy expenditure (EE) at rest, there was no difference between EXE+CAP and EXE (Figure 2b). However, interestingly, CAP generally showed the lowest level of EE throughout the 24 h of analysis compared to the other groups (Figure 2a); additionally, the total EE of CAP tended to be lower than that of CON by 9% (CON: 333.3 ± 40.3, CAP 303.2 ± 26.2 kcal/kg/24 h) (Figure 2b). There was a significant difference between EXE+CAP and CAP but not between EXE and CON.

Similarly, from the results of fat oxidation (FO), CAP generally showed the lowest level of FO (Figure 2c); moreover, the total FO of CAP tended to be lower than that of CON, though it was not significantly different (Figure 2d). From the results of carbohydrate oxidation (CO), there was a significant difference only between the total CO of EXE+CAP and CAP (Figure 2e,f).

Collectively, the results indicated that the additive antiobesity effects of ET with CI did not result from RMR. However, it was considered that the increase of the abdominal fat rate in CAP resulted from the decreased EE at rest.

### 2.5. Correlation between Resting EE and the Proportion of Total Abdominal Fat

To support this assertion, the correlation between EE and the proportion of total abdominal fat was determined. When the analysis was conducted on all the groups, there was a significant, negative, weak correlation (*r* = −0.392; *p* = 0.039) between EE and the proportion of total abdominal fat (Figure 3a). However, there was a significant, negative, moderate to strong correlation (*r* = −0.608; *p* = 0.021) when the analysis was limited to the sedentary groups (Figure 3b). Furthermore, while the values of CAP tended to be concentrated on the upper-left side (lower EE and higher proportion of total abdominal fat), those of CON tended to be concentrated on the bottom-right side (higher EE and lower proportion of total abdominal fat). However, there was no significant correlation (*r* = 0.067; *p* = 0.820) when the analysis was conducted only on the trained groups (Figure 3c).

### 2.6. Metabolism-Associated Protein Expression in Soleus Muscle

The soleus muscle is a typical oxidative skeletal muscle. Citrate synthase (CS) and malate dehydrogenase 2 (MDH2), the key enzymes of oxidative energy metabolism, and peroxisome proliferator-activated gamma coactivator 1-alpha (PGC1α), the transcriptional factor of mitochondrial biogenesis, were analyzed by Western blotting. Additionally, beta-2 adrenoceptor (β2AR), the receptor of adrenaline and noradrenaline predominantly distributed in skeletal muscle, was analyzed as capsiate mainly acts by stimulating the SNS.

While ET significantly elevated the protein expression of CS, MDH2, and PGC1α in the soleus muscle, CI had no effect (Figure 4a–c,e). The protein expression level of β2AR was not affected by ET as well as by CI (Figure 4d,e). Thus, ET with CI did not additively affect metabolism-associated protein expression in soleus muscle.

### 2.7. Metabolism-Associated Protein Expression in Plantaris Muscle

The plantaris muscle is a typical glycolytic skeletal muscle. Thus, hexokinase 2 (HXK2), the rate-limiting enzyme of glycolysis, glucose transporter 4 (GLUT4), and β2AR were analyzed by Western blot analysis.

In line with the results of the soleus muscle, while ET significantly elevated the protein expression of HXK2 and GLUT4 in plantaris muscle, CI had no effect (Figure 5a,b,d). The protein expression level of β2AR was not affected by ET as well as CI (Figure 5c,d). These results revealed that ET with CI did not additively affect the metabolism-associated protein expression in plantaris muscle as well.

### 2.8. Protein Expression of Beta-3 Adrenoceptor in Epididymal Adipose Tissue

Unlike the results of the skeletal muscles, the protein expression level of beta-3 adrenoceptor (β3AR), the receptor of adrenaline and noradrenaline predominantly distributed in the adipose tissues, was solely elevated by the combination of exercise and capsiate (Figure 6a,b). It indicated that the ability of lipolysis was enhanced. Therefore, it was considered that the additive antiobesity effects of ET with CI resulted from the enhancement of lipolysis ability according to β3AR upregulation in abdominal adipose tissue.

## 3. Discussion

The purpose of this study was to examine whether the antiobesity effects of ET can be further enhanced by CI through the modification of energy metabolism in the diet-induced obese mice. Additionally, we examined its effects on the metabolism-associated molecules and key enzymes in skeletal muscles, adipose tissues, and blood. After the 8-week experimental period, the combination of ET and CI exhibited additive effects in preventing obesity. The proportion of abdominal fat and the level of blood leptin were reduced the most by these chronic cotreatments. However, the resting EE was not enhanced. Thus, we assumed that the additive antiobesity effects of ET and CI were due to another cause, and not from RMR. Then, we deduced that these effects may result from a change of metabolism during exercise.

One important feature of ET, related to obesity management, is that it can enhance FO capacity during exercise [4,29]. It means that fat is preferentially oxidized over carbohydrate, as a fuel, even when an equal amount of exercise is performed, at an equal absolute exercise intensity [30]. Similarly, CI before exercise could play a role like a “warm up” to accelerate the FO during exercise. In previous studies, CI was found to activate the SNS [21]. Subsequently, the adrenaline and noradrenaline levels in blood were enhanced by acute treatment with capsiate in humans [22] and rodents [21]; consequently, the released catecholamines promoted fat utilization and thermogenesis [31,32]. Indeed, CI before exercise decreased the respiratory exchange rate and increased fax oxidation during exercise [33]; moreover, according to the results of the current study, the β3AR protein expression of EXE+CAP was significantly upregulated in adipose tissue. This indicated that the ability of lipolysis was enhanced. Additionally, it has been clearly shown that certain bioactive supplements, especially caffeine and green tea extract, when consumed before exercise, enhance FO during exercise [34]. Thus, the additive antiobesity effects of ET and CI may have resulted from an increase in FO during ET.

Unfortunately, there was no additional upregulation of the protein expression of energy metabolism-related key enzymes and factors in the skeletal muscles. Considering previous studies, mild-intensity endurance ET was enough to enhance the energy-metabolic proteins in the skeletal muscles [35]. However, there is no previous study to investigate the effect of capsiate on metabolism-related factors in skeletal muscles. Thus, we deduced that the reason the metabolic enzymes and factors in skeletal muscle were not affected, even by long-term CI, might be associated with the way capsiate functions.

Capsiate-induced EE is mainly caused by the activation of SNS [21], adrenaline secretion from the adrenal medulla [21], and stimulation of nonshivering thermogenesis though uncoupling protein 1 (UCP1) in brown adipose tissue (BAT) [20,36]. As a result, UCP1 protein expression in BAT is subsequently increased by CI [36]. Similarly, the current study showed that the level of blood noradrenaline was upregulated in CAP and UCP1 protein expression in BAT was increased by CI in both the sedentary and exercise-trained groups (*p* = 0.054) (Appendix B, Figure A1a,b). However, the metabolism-related proteins in skeletal muscles did not show any differences or even any trends between the untreated and capsiate groups. Hence, it can be deduced that skeletal muscles cannot be the main target of capsiate in increasing EE. Therefore, we suggest that further studies on the effect of capsiate on skeletal muscles will be required to confirm whether another metabolism-related molecular pathway, such as 5′ AMP-activated protein kinase (AMPK) signaling pathway, is associated or not.

In the current study, there were certain unexpected results such as CI without exercise increased the proportion of total abdominal fat in diet-induced obese mice. This result is contrary to that of most previous studies that reported the antiobesity effects of capsiate [15,22,24]. However, some previous studies on capsiate concur with our results. A critical review and meta-analysis of studies on capsiate [37] reported that four out of seven studies that evaluated the effect of capsiate on weight loss in humans found it to be effective, whereas the other three studies reported no effect. The impact of capsiate was more likely to be negative when the subjects of the experiment were mainly overweight and/or obese. Thus, the author of this review suggested that further investigation was required in overweight/obese individuals to confirm whether capsiate is an appropriate compound for prevention or treatment of obesity. In this perspective, given that the subjects of the current study were mice fed with high-fat diet, our finding that chronic intake of capsiate could increase the proportion of total abdominal fat is worthy of attention. The fact that the blood leptin concentration was elevated in CAP could support this result. Leptins are released from adipocytes, hence, as obesity progresses more due to high-fat diet, the blood leptin level is gradually elevated [38,39].

From the results of our study, we deduced that the increased abdominal fat rate by chronic CI resulted from a decreased EE at rest, although there have been no previous reports of capsiate reducing EE at rest. It is also partially supported by the negative correlation between EE and the proportion of total abdominal fat. To explain these unexpected results, we tentatively suggest that there are two possible pathways: (1) capsiate affects the central nervous system, which may be related to the decreased EE at rest and (2) sympathetic hyperactivity could decrease the ability to dissipate calories by the downregulation of βAR.

In a previous study that used functional magnetic resonance imaging (fMRI), several regions of the brain, including the hippocampus, amygdala, thalamic nuclei, and hypothalamic areas, were affected by the intragastric infusion of capsiate via the capsaicin and capsiate receptors (transient receptor potential cation channel subfamily V member 1, TRPV1) located in the central nervous system [40]. Among the affected regions, the hippocampus is closely associated with depression-like behavior [41,42], and depression could cause obesity by biologically, psychologically, and behaviorally attenuating metabolism [43,44]. Additionally, it has been well documented that depressive symptoms and the level of physical activity are negatively correlated [44,45]. Interestingly, during the progression of the current study, we observed that the sedentary groups, especially CAP, were relatively inactive while the exercise-trained groups were not. Thus, after careful consideration, we decided to conduct the tail suspension test (TST), a method of assessing the level of depression in rodents [46], to estimate the level of physical activity, using the correlation between immobility time (measured by TST) and EE at rest. Notably, the immobility time of CAP was significantly longer than that of CON but there was no difference between EXE and EXE+CAP (Appendix C, Figure A2a). Furthermore, there was a significant, negative, weak to moderate correlation (*r* = −0.417, *p* = 0.047) between the immobility time and EE (Figure A2b). Additionally, the values of CAP tended to be concentrated on the upper-left side (lower EE and longer immobility time) in contrast to the other groups.

While there was no report of the effect of capsiate on the hippocampus, there was evidence that capsaicin can attenuate hippocampal function. In vivo, proliferation of neural progenitor cells in the hippocampus was significantly suppressed by intraperitoneal administration of capsaicin for 2 weeks [47]. In vitro, the Notch and Hedgehog pathways, which inhibit neural progenitor cell proliferation in the hippocampus, were activated by capsaicin through TRPV1, which resulted in the attenuated proliferation of hippocampal neural progenitor cells [48]. Considering these previous findings, we assumed that chronic intake of capsiate might induce depression-like behavior by suppressing hippocampal neurogenesis, with the result that mice became inactive and then liable to gain weight. Additionally, given that exercise is a well-known treatment to improve hippocampal neurogenesis [49,50], the fact that the characteristic features of CAP did not occur in EXE+CAP could provide supporting evidence.

The other possible mechanism is related to sympathetic hyperactivity. A previous study asserted that sympathetic hyperactivity might decrease the ability to dissipate calories, via downregulation of βAR [51]. In another study using isoproterenol, an agonist of the βAR, heart rate and EE responses were less sensitive to infusions of isoproterenol in a sympathetic-hyperactivated group than in a control group. Furthermore, the heart rate and EE responses were negatively correlated with the norepinephrine levels, indicating that sympathetic hyperactivity caused a downregulation of βAR responses [52].

Intake of capsiate activates the SNS [20] causing subsequent rise in the blood concentration of noradrenaline [22]. Sympathetic nerves are also activated by leptin released from adipocytes [53], and a person who has a higher body mass index or fat mass also has a higher level of plasma leptin [54] owing to obesity-induced leptin resistance [39]. In these contexts, it is very likely that the SNS was hyperactivated in the CI groups owing to the intake of both capsiate and high-fat diet. Indeed, in the current study, the blood levels of noradrenaline and leptin were most significantly increased in CAP.

Considering all these points, the reduction in resting EE in CAP might be caused by sympathetic hyperactivation via downregulation of βAR. Additionally, given that the enhancement of parasympathetic regulation is one of the responses to ET [55,56], the results that the blood noradrenaline level was not increased in EXE+CAP and the resting EE was not decreased in EXE+CAP could provide supporting evidence. However, βAR protein expression levels of CAP were not affected in skeletal muscles and adipose tissue. Thus, further studies on the current topic are recommended with a focus on metabolism and SNS.

The major limitation of this study is that the evidence at molecular level was insufficient to validate the two main findings of the study. However, we tried to show various results to support our assertions. Additionally, we discussed our main findings in a broad context so that they can provide sufficient insights for future studies. Second, a naïve group was excluded from our study. Considering that feeding the high-fat diet is a verified method of inducing obesity in rodent models [57] and our main interest was to examine the antiobesity effects of ET and CI on the obese subjects, it was not necessary to include a naïve group. Third, we cannot calculate sample size to determine the appropriate number of animals because there were not enough suitable previous studies to estimate Cohen’s f.

## 4. Materials and Methods

### 4.1. Animal Care

Before the experimental period, 8-week-old male ICR mice (34.9 ± 1.92 g) were adapted to the laboratory environment. The mice were purchased from Orient Bio Inc., Seongnam, Korea. All the mice (*n* = 32) were housed in standard plastic cages (four mice per cage) under controlled humidity (50–55%), temperature (23 ± 1 °C), and lighting (12:12-h light-dark cycle; lights on at 08:00). They were fed a high-fat diet, where 60% of the energy intake was from fat (HFD; Research Diets, Inc., New Brunswick, NJ, USA) (Appendix A), and water was available ad libitum. This study was approved by Konkuk University Institutional Animal Care and Use Committee (No. KU18005-3; 2018-11-08).

### 4.2. Study Design

Mice were randomly divided into four groups (*n* = 8 per group): CON, CAP, EXE, and EXE+CAP. CAP and EXE+CAP were orally administrated 10 mg/kg of capsiate. CON and EXE were also orally administrated the equal volume of solvent without capsiate. BW and FI were measured daily.

ET intensity was set at “mild” (about 60% maximal oxygen uptake), to prevent any concealment of the capsiate effect that might arise from harder training. The training by treadmill was conducted on 09:00–10:00, five times per week for 8 weeks and the absolute exercise intensity was gradually increased to avoid decreasing the relative exercise intensity (Appendix A). Capsiate was administered 30 min before training (Figure 7).

### 4.3. RMR Analysis

To investigate the additive effects of ET with CI on energy metabolism at rest, RMR analysis was conducted on the last weekend of the experiment, for 24 h. Capsiate was administered just before the measurement. HFD and water were served ad libitum. Mice were isolated and randomly assigned to metabolic chambers to measure the energy metabolism at rest. The metabolic chambers use an open-circuit method and the volume of each chamber was 3 L. The average flow rate for each chamber was set to 3 L/min. An acrylic tube was connected to each chamber for the manipulation of air volume. Respiratory gas analysis (O_2_ uptake and CO_2_ production) was conducted by using a mass spectrometer (ALCO-2000, ARCO System, Chiba, Japan) via a switching system (ARCO-2000-GS-8, ARCO System, Chiba, Japan) that allowed the spectrometer to sample the gas from each chamber, every 15 s in turn. EE, FO, and CO were calculated from the measurements of respiratory gas.

### 4.4. Tissue Preparation

At the end of the experiment, all the mice were restricted from intake of food for 2 h before dissection, to control variation induced by food mass intake. BAT, soleus muscles, plantaris muscles, epididymal, perirenal, and mesenteric fat were surgically obtained from the mice, under deep anesthesia from 10 μL/g of 1.25% avertin. Total abdominal fat was calculated from the sum of epididymal, perirenal, and mesenteric fat. All the tissues were weighed and stored at −80 °C immediately after dissection.

Note that the data of BW, FI, and abdominal fat, the result of Table 1, were coused with the previously published paper [33].

### 4.5. Blood Parameters

The venous blood samples were immediately collected and allowed to clot for 30 min at 24–25 °C before centrifuging at 2000× *g* for 15 min at 4 °C. Serum was transferred to a new tube and stored at −80 °C. We used ELISA kits for the analysis of the serum concentrations of leptin (KMC2281, Thermo Fisher Scientific, Waltham, MA, USA), noradrenaline (CSB-E07870m, CUSABIO, Wuhan, China), and insulin (80-INSMS-E01, ALPCO, Salem, NH, USA) and colorimetric kit for the detection of the serum concentration of glucose (K039-H1, Arbor Assays, Ann Arbor, MI, USA).

### 4.6. Western Blot Analysis

The right lobe of BAT, right soleus muscle, right plantaris muscle, and epididymal fat (150 mg) were homogenized using a homogenizer with 500, 180, 250, and 230 μL of protein extraction buffer, respectively (EzRIPA Lysis kit, ATTO, Tokyo, Japan). For obtaining adipose tissue samples, lysates were centrifuged at 20,000× *g*, 4 °C for 15 min. Then, the layer of lipids (top layer) was removed, and clear supernatants were transferred to a new tube. The supernatants were centrifuged again at 20,000× *g*, 4 °C for 15 min. Finally, the supernatants were transferred to a new tube. For samples from other tissues, lysates were centrifuged at 14,000× *g*, 4 °C for 15 min and the supernatants were transferred to a new tube. The protein concentration of the lysates was determined by BCA assay kit (Thermo Fisher Scientific, Waltham, MA, USA). BAT and skeletal muscle samples were denatured by heating at 100 °C for 5 min whereas the adipose tissues were not denatured. Total protein (20 μg per lane) was separated by 10% SDS-PAGE at 45 V for 15 min, 65 V for 15 min, and 100 V for 120 min continuously, and transferred to polyvinylidene difluoride membranes (Millipore, Billerica, MA, USA) at 100 V for 90 min. The membranes were blocked for 1 h at 24–25 °C with phosphate-buffered saline (PBS) containing 5% skim milk (DB Difco, Franklin Lakes, NJ, USA) and then washed four times for 5 min each using PBS with 0.1% Tween 20 (PBS-T). After overnight incubation at 4 °C with primary antibodies with PBS-T containing 3% skim milk, the membranes were washed with PBS-T and incubated with a horseradish peroxidase-conjugated secondary antibody with PBS-T containing 3% skim milk for 1 h at 24–25 °C (information of antibodies is provided in the Appendix A). Immunodetection was carried out with ECL reagent (Amersham Biosciences, Uppsala, Sweden). All images showing the results of quantitative analysis were assessed using Image J software (NIH Image Engineering, Bethesda, MD, USA).

### 4.7. Statistical Analysis

All data were analyzed by the IBM SPSS Statistics 25 software. Significant differences in the means were determined using a two-way analysis of variance (ANOVA), followed by the LSD post hoc test, which was used to determine differences in the group means. Significant differences in value over time were determined using a two-way repeated ANOVA. Significant correlation between EE at rest and total abdominal fat were determined using Pearson’s correlation analysis. Values of *p* < 0.05 were considered statistically significant and all the results are presented as mean ± standard deviation.

## 5. Conclusions

In summary, the purpose of this study was to examine whether the antiobesity effects of ET can be further enhanced by CI via change in resting energy metabolism and to determine its effects on metabolic molecules in diet-induced obese mice. After the 8-week experimental period, ET combined with CI additively reduced abdominal fat rate compared to ET alone, but the resting EE was not affected by CI in the trained group. Thus, we assumed that CI before exercise may enhance FO during exercise and consequently cause loss of fat. However, the chronic intake of capsiate without exercise increased the abdominal fat rate in obesity-induced mice, which may result from the reduced resting EE. Then, we tentatively suggested that the reduced resting EE may be attributed to the following reasons: (1) capsiate might affect activity levels via suppression of hippocampal function and (2) the combination of capsiate and high-fat diet could downregulate βAR via sympathetic hyperactivation. Therefore, we suggest that capsiate could be a candidate supplement for maximizing the antiobesity effects of ET via upregulation of β3AR in adipose tissue. However, capsiate treatment without exercise during high-fat diet may not be beneficial in managing obesity (Figure 8).

## Figures and Tables

**Figure 1 ijms-22-00769-f001:**
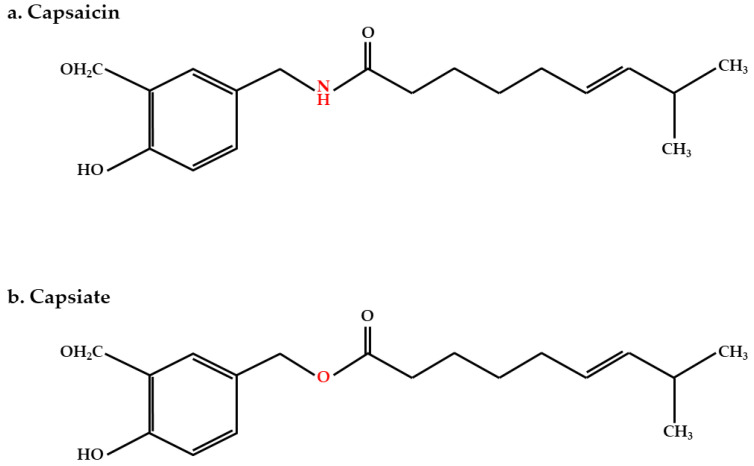
Chemical structures of capsaicin (**a**) and capsiate (**b**).

**Figure 2 ijms-22-00769-f002:**
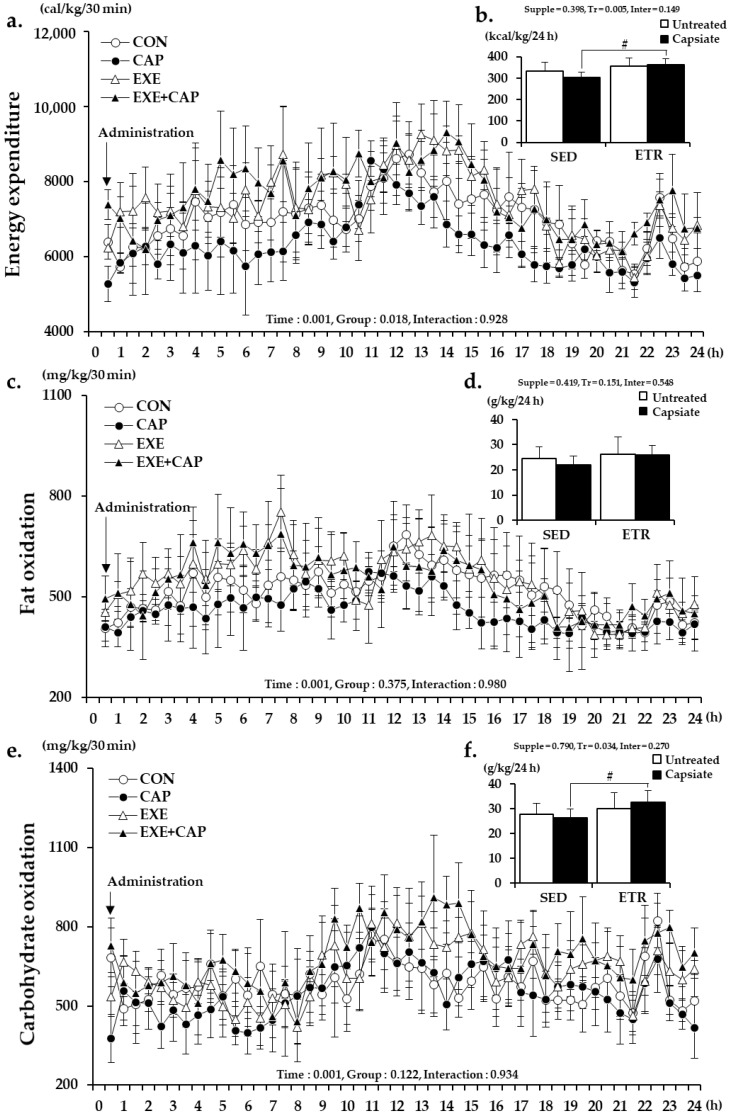
RMR data for EE, FO, and CO: (**a**,**c**,**e**) represent the changes in EE, FO, and CO over time for 24 h, respectively; (**b**,**d**,**f**) represent the total EE, FO, and CO for 24 h, respectively. Supple, supplement effect; Tr, training effect; Inter, interaction effect; RMR, resting metabolic rate; EE, energy expenditure; FO, fat oxidation; CO, carbohydrate oxidation; CON, sedentary control group; CAP, sedentary with capsiate intake group; EXE, exercise-trained control group; EXE+CAP, exercise-trained with capsiate intake group; SED, sedentary groups; ETR, exercise-trained groups. Values represent the mean ± standard deviation (*n* = 7). ^#^
*p* < 0.05.

**Figure 3 ijms-22-00769-f003:**
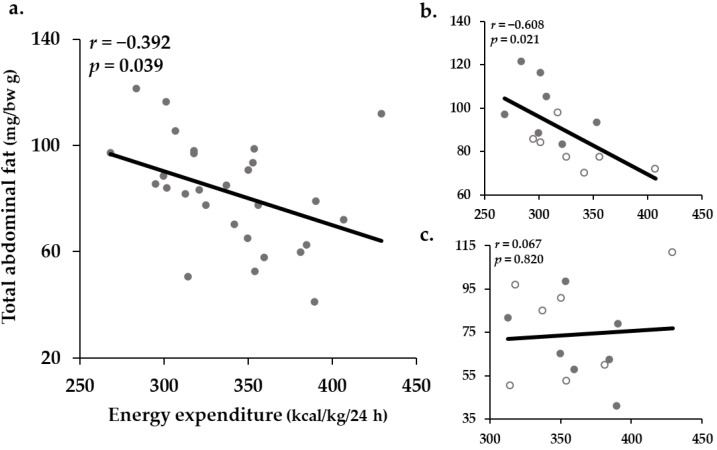
The results of correlation between resting energy expenditure and the proportion of total abdominal fat: (**a**) graph indicating correlation with all the groups; (**b**) graph indicating correlation with the sedentary groups; (**c**) graph indicating correlation with the exercise trained groups. In (**b**,**c**) empty dots represent the untreated group. (*n* = 7 per group).

**Figure 4 ijms-22-00769-f004:**
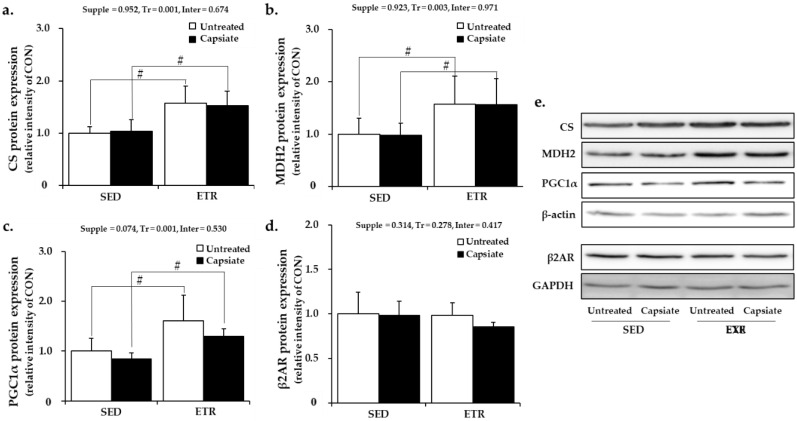
Results of metabolism-associated protein expression in soleus muscle: (**a**–**d**) quantification of CS, MDH2, PGC1α, and β2AR, respectively; (**e**) representative Western blot images. Supple, supplement effect; Tr, training effect; Inter, interaction effect; CS, citrate synthase; MDH2, malate dehydrogenase 2; PGC1α, peroxisome proliferator-activated gamma coactivator 1-alpha; β2AR, beta-2 adrenoceptor; GAPDH, glyceraldehyde 3-phosphate dehydrogenase; CON, sedentary control group; SED, sedentary groups; ETR, exercise-trained groups. Values represent the mean ± standard deviation (*n* = 6). ^#^
*p* < 0.05.

**Figure 5 ijms-22-00769-f005:**
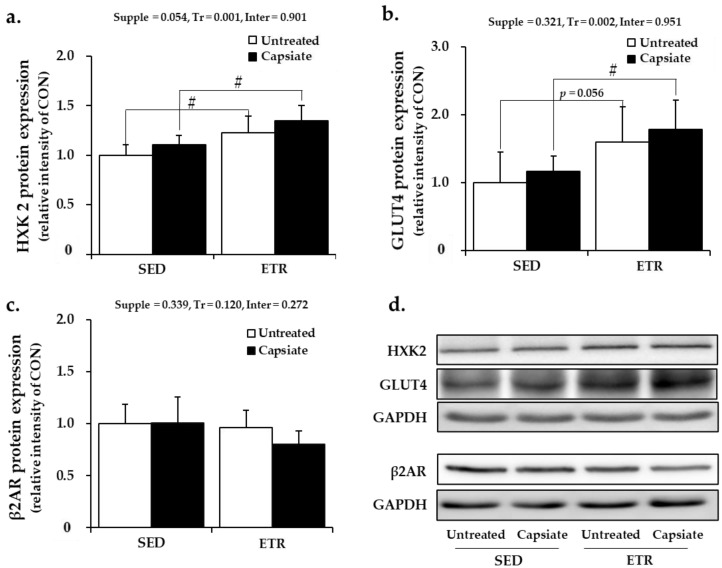
Results of metabolism-associated protein expression in plantaris muscle: (**a**–**c**) quantification of HXK2, GLUT4, and β2AR, respectively; (**d**) the representative Western blot images. Supple, supplement effect; Tr, training effect; Inter, interaction effect; HXK2, hexokinase 2; GLUT4, glucose transporter 4; β2AR, beta-2 adrenoceptor; GAPDH, glyceraldehyde 3-phosphate dehydrogenase; CON, sedentary control group; SED, sedentary groups; ETR, exercise-trained groups. Values represent the mean ± standard deviation (*n* = 6). ^#^
*p* < 0.05.

**Figure 6 ijms-22-00769-f006:**
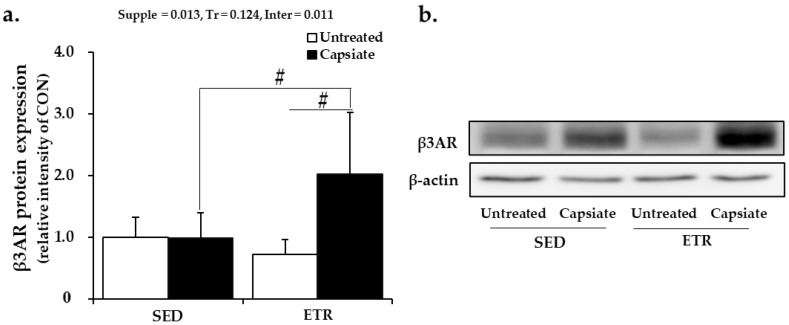
Results of beta-3 adrenoceptor protein expression in epididymal adipose tissue: (**a**) quantification of β3AR; (**b**) the representative Western blot images. Supple, supplement effect; Tr, training effect; Inter, interaction effect; β3AR, beta-3 adrenoceptor; CON, sedentary control group; SED, sedentary groups; ETR, exercise-trained groups. Values represent the mean ± standard deviation (*n* = 6). ^#^
*p* < 0.05.

**Figure 7 ijms-22-00769-f007:**
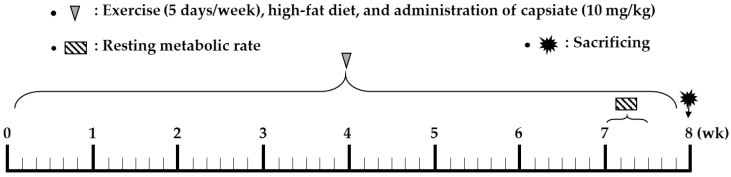
Study design.

**Figure 8 ijms-22-00769-f008:**
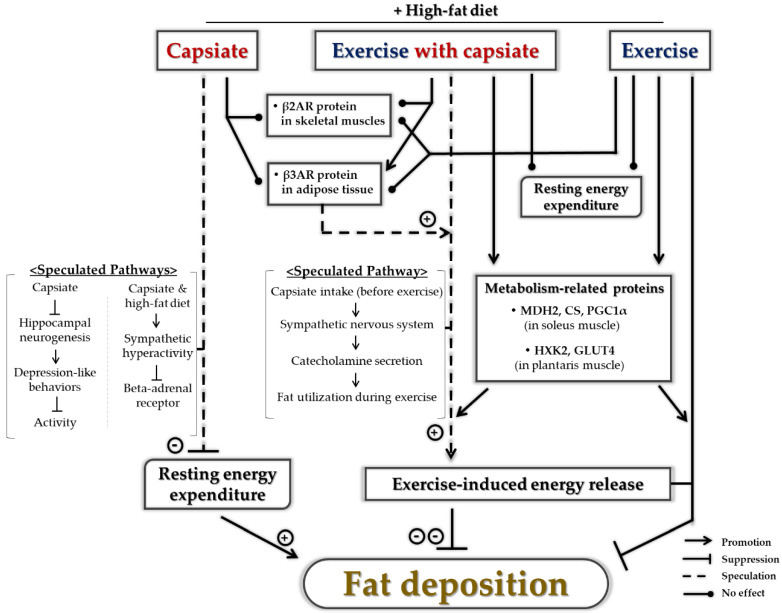
Flow diagram showing summary of the current study. β2AR, beta-2 adrenoceptor; β3AR, beta-3 adrenoceptor; MDH2, malate dehydrogenase 2; CS, citrate synthase; PGC1α, peroxisome proliferator-activated gamma coactivator 1-alpha; HXK2, hexokinase 2; GLUT4, glucose transporter 4.

**Table 1 ijms-22-00769-t001:** Results of body weight, body weight gain, food intake, feed efficiency ratio, and the proportion of adipose tissue.

		CON	CAP	EXE	EXE+CAP
BW (g)	Pre	34.91 ± 1.87	35.08 ± 2.49	35.28 ± 1.84	34.69 ± 1.48
	Post	47.79 ± 1.54	49.40 ± 4.95	46.95 ± 4.70	44.03 ± 1.22 ^†^
BWG (g/8 wk)		12.88 ± 2.36	14.32 ± 2.63	11.68 ± 4.51	9.34 ± 1.15 ^†^
FI (g/8 wk/mouse)		177.7 ± 3.00	169.7 ± 12.4	188.5 ± 11.1 *	201.5 ± 3.96 ^†,§^
FER (BWG/FI*100)		7.26 ± 1.42	8.49 ± 1.75	6.21 ± 2.36	4.64 ± 0.61 ^†^
Adipose tissue (mg/BW g)					
Epididymal		37.33 ± 4.74	46.93 ± 6.64 *	33.50 ± 9.11	28.53 ± 8.70 ^†^
Perirenal		15.34 ± 3.99	20.14 ± 4.25 *	16.71 ± 6.53	11.11 ± 2.88 ^†^
Mesenteric		27.40 ± 3.21	30.85 ± 3.85	25.09 ± 2.92	22.58 ± 3.65 ^†^
Total		80.07 ± 7.15	97.92 ± 10.2 *	75.30 ± 18.1	62.21 ± 14.6 ^†^

CON, sedentary control group; CAP, sedentary with capsiate intake group; EXE, exercise-trained control group; EXE+CAP, exercise-trained with capsiate intake group; BW, body weight; BWG, body weight gain; FI, food intake; FER, feed efficiency ratio. Values represent the mean ± standard deviation (*n* = 8). * *p* < 0.05 vs. CON; ^†^
*p* < 0.05 vs. CAP; ^§^
*p* < 0.05 vs. EXE.

**Table 2 ijms-22-00769-t002:** Leptin, noradrenaline, glucose, and insulin concentrations in blood.

	CON	CAP	EXE	EXE+CAP
Leptin(ng/mL)	18.97 ± 4.42	26.83 ± 9.12 *	18.51 ± 8.95	9.84 ± 5.74 ^†,§^
Noradrenaline (pg/mL)	73.54 ± 9.32	86.87 ± 5.02 *	80.45 ± 2.89	78.17 ± 4.01 ^†^
Glucose (mg/dl)	153.3 ± 46.8	172.3 ± 47.1	194.6 ± 66.1	186.0 ± 66.1
Insulin (ng/mL)	0.86 ± 0.24	0.83 ± 0.28	0.77 ± 0.32	0.79 ± 0.39

CON, sedentary control group; CAP, sedentary with capsiate intake group; EXE, exercise-trained control group; EXE+CAP, exercise-trained with capsiate intake group. Values represent the mean ± standard deviation (*n* = 8). * *p* < 0.05 vs. CON; ^†^
*p* < 0.05 vs. CAP; ^§^
*p* < 0.05 vs. EXE.

## Data Availability

The data presented in this study are available in insert article and Appendix A here and on request from the corresponding author.

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
