# Peer review of "Capsiate Intake with Exercise Training Additively Reduces Fat Deposition in Mice on a High-Fat Diet, but Not without Exercise Training"

_ijms, 2021, doi:10.3390/ijms22020769_

Round 1

Reviewer 1 Report

The manuscript is well-written and provides sound scientific rationale of the study. 

Below are some comments that can improve the manuscript further:

  1. Please explain rationale for using male mice. Do sex-specific hormones or body composition plays a role in this study?
  2. Specify the duration of ET during each session. Was it done during the day or night when mice are most active?
  3. If tissues are available, please show a picture of adipose tissue in control vs treatment groups for comparison

Author Response

<Responses to Reviewer 1.>

Thank you very much for your constructive comments and suggestions. We have considered all the comments carefully and revised our manuscript accordingly. Our responses to each comment (shown in Italics) are as follows.

Question 1. Please explain rationale for using male mice. Do sex-specific hormones or body composition plays a role in this study?

Response: In the field of metabolic disease and obesity, when it comes to animal study, male mice have been commonly selected because male mice exhibit more pronounced obesity-related phenotypes than female mice especially when obesity is induced with high-fat diet. Also, male mice are free from the metabolic variability that can be induced in female mice by the estrous cycle. These common reasons were our fundamental rationale for using male mice. However, there was another important reason. From several previous human studies of capsiate, there was no different effect depending on sex. Then we regarded capsiate as a compound that is relatively free from the sex-differences matter for using on metabolic experiment. Considering all these points, it was more appropriate to use male mice rather than female for control of experimental condition to achieve our purpose.

Question 2. Specify the duration of ET during each session. Was it done during the day or night when mice are most active?

Response: I think we gave you confusion about the information of ET session due to the location of citation of figure 7 (line 346 in the pre-revised manuscript). Thus, we changed the location of it to next to last sentence of subsection 4.2. (line 351). The detailed ET protocol was submitted as supplementary material (Table S3) to improve readability of paper because the ET protocol was rather not simple to write in sentence. Please refer to line 350 that we highlighted for the response and Table S3.

Every ET was done on 09:00-10:00 AM. This period is a beginning phase for mouse to be changed to inactive statement from active statement by their circadian rhythm. However, this period is when mice are the most active among the available times under experimental environment because light turns on at 08:00 AM. Many investigators choice this period when they conduct ET experiment. Therefore, our laboratory have been using the protocol that ET is conducted on 09:00-10:00 AM and have been publishing many papers using this protocol (please refer to the articles: https://doi.org/10.1186/s12986-019-0406-z, http://dx.doi.org/10.20463/jenb.2017.0056, https://doi.org/10.1186/1550-2783-11-35).

According to your comment, we added the information of time on which ET was performed. Please refer to line 349 that we highlighted for the response.

Question 3. If tissues are available, please show a picture of adipose tissue in control vs treatment groups for comparison.

Response: Please excuse that the tissues are not available because most parts of them were used or preprocessed for biochemical analysis. So, intact tissues suitable to objectively represent each group are no left.

Reviewer 2 Report

The article focuses on the role of Capsiate in the enhancing effects of weight loss. The authors used the correct research methods. Adequate conclusions were made based on the results of the study. The article is of interest to a wide range of researchers.

Authors should check some typos and graphics before publishing the article.

Author Response

<Responses to Reviewer 2.>

Thank you very much for your constructive comments and suggestions. We have considered all the comments carefully and revised our manuscript accordingly. Our responses to each comment (shown in Italics) are as follows.

Question 1. Check some typos and graphics.

Response: We deleted the typos of repeated abbreviation. Please refer to line 125 that we highlighted for the response.

Before: 2.4. Resting metabolic rate (RMR)

After: 2.4. Resting metabolic rate

- The mark of statistical significance was larger to improve visibility. Please refer to line 184 and 192 that we highlighted for the response.

- We inserted “g”, unit of weight. Please refer to line 335 that we highlighted for the response.

Reviewer 3 Report

I have the following comments to make:

  1. The authors should clearly state what the novelty of their study is. It comes as a surprise why authors chose to divide the presentation of their results into 2 papers, as they both refer to the same experiment (with the addition of one more experimental group): the manuscript that was submitted in the IJMS and a very recently published paper [Hwang D, Seo JB, Kim J, Lim K. Effect of mild-intensity exercise training with capsiate intake on fat deposition and substrate utilization during exercise in diet-induced obese mice. Phys Act Nutr. 2020 Sep;24(3):1-6.]. Please comment on that.
  2. Even though some previously published information may be necessary to be included in the present submission -for the full and better comprehension of the subject-, the authors are strongly recommended to acknowledge which results have been shown in their previously published paper in order to avoid duplicate publication. For example, paragraphs 2.1 and 2.2 of the results section as well as the results described in the abstract regarding abdominal fat proportion are among them. The title of the article has also to be simplified and focused on the novel findings of the present submission. The aim of the study has to be presented in one paragraph: “… the objective of this study was to examine whether the anti-obesity effects of ET can be further enhanced by CI through a change in energy metabolism in diet-induced obese mice…. Furthermore, we aimed to explore the effects of the combination treatment on metabolism-associated molecules and key enzymes in the blood, adipose tissue, and skeletal muscles….”. Ideally, the authors should also paraphrase parts in the present submission that have also been included in their previous published paper.
  3. The methods section has to be more detailed. For example the molecules that are determined through western blot analysis have to be included in the corresponding paragraph (4.6). I would also suggest that the authors give some extra information about the animals used in their experiments (ICR).
  4. Did the author make sample size calculations to determine the appropriate number of animals needed to answer their research question? Please clarify and include this information either in the methods section or as limitation of the study.
  5. Some of the experimental details have to be included in the abstract as well, including the number of the groups and the number of the animals.
  6. The authors are suggested to clarify their results presented in tables as far as the level of statistical significance is concerned. In other words, when statistical significance is mentioned in this submission, it is not clear based on both the figure and the description of the legend which groups are compared each time.
  7. As the number of 68 references is great enough, the authors are suggested to try to decrease the total number of references by selecting those references that are more recent, relevant and significant. They should also update their reference list in terms of including available relative articles that describe findings from similar experimental approaches as well.
  8. Few grammar/syntax mistakes exist throughout the text. The authors are suggested to revise their manuscript ideally with the help of a native English speaker.

Round 2

Reviewer 3 Report

Comments sufficiently answered.